# Bioimpedance Vector References Need to Be Period-Specific for Assessing Body Composition and Cellular Health in Elite Soccer Players: A Brief Report

**DOI:** 10.3390/jfmk5040073

**Published:** 2020-10-01

**Authors:** Tindaro Bongiovanni, Gabriele Mascherini, Federico Genovesi, Giulio Pasta, Fedon Marcello Iaia, Athos Trecroci, Marco Ventimiglia, Giampietro Alberti, Francesco Campa

**Affiliations:** 1Department of Health, Performance and Recovery, Parma Calcio 1913, 40121 Parma, Italy; tindaro.bongiovanni@gmail.com; 2Department of Biomedical Sciences for Health, Università degli Studi di Milano, 20129 Milano, Italy; marcello.iaia@unimi.it (F.M.I.); Athos.Trecroci@unimi.it (A.T.); gianpietro.alberti@unimi.it (G.A.); 3Department of Experimental and Clinical Medicine, Università degli Studi di Firenze, 50139 Florence, Italy; 4Medical Department Manchester City Football Club, Manchester 03101, UK; fede.genovesi@libero.it; 5Medical Department Parma Calcio 1913, 40121 Parma, Italy; ghitopasta@hotmail.com; 6Inflammatory Bowel Disease Unit, A.O.O.R. Villa Sofia-Cervello, 90146 Palermo, Italy; marco.ventimiglia@unipa.it; 7Department for Life Quality Studies, University of Bologna, 47921 Rimini, Italy; francesco.campa3@unibo.it

**Keywords:** BIVA, phase angle, R-Xc graph, tolerance ellipses

## Abstract

Purpose: Bioimpedance data through bioimpedance vector analysis (BIVA) is used to evaluate cellular function and body fluid content. This study aimed to (i) identify whether BIVA patters differ according to the competitive period and (ii) provide specific references for assessing bioelectric properties at the start of the season in male elite soccer players. Methods: The study included 131 male soccer players (age: 25.1 ± 4.7 yr, height: 183.4 ± 6.1 cm, weight: 79.3 ± 6.6) registered in the first Italian soccer division (Serie A). Bioimpedance analysis was performed just before the start of the competitive season and BIVA was applied. In order to verify the need for period-specific references, bioelectrical values measured at the start of the season were compared to the reference values for the male elite soccer player population. Results: The results of the two-sample Hotelling T^2^ tests showed that in the bivariate interpretation of the raw bioimpedance parameters (resistance (R) and reactance (Xc)) the bioelectric properties significantly (T^2^ = 15.3, F = 7.6, *p* ≤ 0.001, Mahalanobis D = 0.45) differ between the two phases of the competition analyzed. In particular, the mean impedance vector is more displaced to the left into the R-Xc graph at the beginning of the season than in the first half of the championship. Conclusions: For an accurate evaluation of body composition and cellular health, the tolerance ellipses displayed by BIVA approach into the R-Xc graph must be period-specific. This study provides new specific tolerance ellipses (R/H: 246 ± 32.1, Xc/H: 34.3 ± 5.1, r: 0.7) for performing BIVA at the beginning of the competitive season in male elite soccer players.

## 1. Introduction

Body composition analysis is currently one of the most studied evaluations in sport, mainly for the relationship between physical characteristics and sports performance [1]. In sports, excess fat mass reduces endurance performance, while an increase in lean mass, especially muscle mass, is associated with an increase in power and strength [2]. Furthermore, the assessment of localized body composition allows the identification of differences in muscle mass and strength between areas of the body and may allow a reduction in the risk of injury (evaluation of contralateral limbs, agonist-antagonists) [3].

Body composition assessment should also be considered in sports involving weight categories, where athletes benefit from being placed in a lower weight category, in these cases any weight loss must therefore be monitored closely. Excessive training coupled with calorie restrictions can lead to excessive, unnecessary and dangerous weight loss. This weight loss in both women and men decreases performance, bone mineral density, muscle mass and is detrimental to health [4,5].

Bioelectrical impedance vector analysis (BIVA) is a method widely used to evaluate body composition and cellular health in athletes, as well as in the general population [6,7,8,9]. This method considers the raw bioelectrical parameters (resistance and reactance) standardized for the height of the subjects as a vector within a graph. Resistance (R) is the opposition to the flow of an injected alternating current, at any current frequency, through intra- and extra-cellular ionic solutions, while reactance (Xc) represents the dielectric or capacitive component of cell membranes and organelles, and tissue interfaces [10].

BIVA allows for the monitoring of vector changes over time or the comparison of the vector position within the R-Xc graph on specific population tolerance ellipses [11,12,13]. Given the ease and repeatability of this method, several references for athletes have recently been proposed, including those for soccer players [14], volleyball players [15], and cyclists [16], while also considering the competitive level of the athlete.

In soccer, Levi Micheli et al. [14] were the first to demonstrate how athletes need to be assessed on specific tolerance ellipses, showing bioelectric values that were far different than those of the normal healthy population. Subsequently, Mascherini et al. [17] suggested how bioimpedance vectors show displacements over the season, reflecting the changes that occur in the body composition and physical condition of the players. This was later confirmed by Campa et al. [18] who analyzed the bioelectrical changes comparing BIVA to results obtained by Dual X-ray Absorptiometry (DXA) and dilution techniques over a season in athletes, also showing that these vector changes occur in many other sports.

During the different phases of competition, the one which precedes the start of the season is among the most important periods in which to evaluate the athlete’s physical condition and the body composition adjustments that are sought during the pre-season. Considering the vector changes that occur over the season, the bioelectrical references used in the BIVA assessment must be specific for the competitive period in which the athlete is tested. Therefore, the purpose of this study was to show how BIVA references provided in different phases of the season differ in male elite soccer players, also providing new references for assessing body composition in the start-of-the season period.

## 2. Materials and Methods

### 2.1. Design and Participants

A total of 131 male professional soccer players (age: 25.1 ± 4.7 yr; height: 183.4 ± 6.1 cm; weight: 79.3 ± 6.6 Kg) were recruited and participated in this observational study.

The inclusion criteria were: (1) players registered and participating in the first (Serie A) Italian National division; (2) non-injured at the time of the assessment. After having been informed about the aims and the procedures of the research, all athletes gave their written informed consent. The project was approved by the Bioethics Committee of the University of Milan (approval code: 1052019) and was conducted in accordance with the guidelines of the declaration of Helsinki.

### 2.2. Procedures

All measurements were performed in resting and fasting conditions at the facilities of the teams in the last week of August at 8.30 a.m. Generally, this period corresponds to the end of the preparation for the competitive season; therefore, it coincides with the start of the season. Body height was recorded to the nearest 0.1 cm with a stadiometer (SECA^®^ 240, Hamburg, Germany) and weight was measured to the nearest 0.1 Kg with a calibrated weight scales (SECA^®^ 877, Hamburg, Germany).

Whole-body impedance was obtained using a bioimpedance analyzer (BIA 101 Anniversary Edition, Akern, Florence, Italy). The device emits an alternating sinusoidal electric current of 400 mA at an operating single frequency of 50 kHz (±0.1%). Subjects were positioned with a leg opening of 45° with respect to the midline of the body, and with the upper limbs positioned 30° away from the trunk. The bioelectric phase angle (PhA) was calculated as the arctangent of Xc/R × 180/π. BIVA was carried out using the classic methods, e.g., normalizing R (ohm) and Xc (ohm) for height in meters [6,8].

### 2.3. Statistical Analyses

The two-sample Hotelling T^2^ test was used to compare the differences in the mean impedance vectors between the bioimpedance data measured on the athletes of this study and the reference bioelectric values proposed by Levi Micheli et al. [14] The 50, 75, and 95% tolerance ellipses were generated using the BIVA software [19]. Statistical significance was predetermined as *p* < 0.05. Data were analyzed with IBM SPSS Statistics, version 24.0 (IBM Corp., Armonk, NY, USA).

## 3. Results

Table 1 shows anthropometric and bioelectrical characteristics of the soccer player.

The results of the two-sample Hotelling’s T^2^ test showed separate 95% confidence ellipses indicating a significant difference (T^2^ = 15.3, F = 7.6, *p* ≤ 0.001, Mahalanobis D = 0.45) between the BIVA patters measured in this study and those proposed by Levi Micheli et al. [14] as a reference for the male elite soccer players population (Figure 1a).

The new reference ellipses and the single bioimpedance vectors measured in the soccer players at the start of the season are shown in Figure 1b.

## 4. Discussion

The aim of this study was to show the importance of evaluating bioelectric properties using BIVA references that are suitable for the competitive period in which the assessment is carried out. The results of this study, which provide bioelectrical impedance data for 131 elite players, showed how the tolerance ellipses created on the basis of measurements during the different phases of the competition differ significantly for elite soccer players.

The bioimpedance data reported in the present study are comparable to previous values reported during the start-of-the season period in elite soccer players [20,21,22]. In comparison with the elite Italian male soccer population investigated by Levi Micheli et al. [14], the elite soccer players measured in this study showed a significant vector shift to the left on the minor axis of the tolerance ellipses. This could indicate a greater cell mass, which is a consequence of the effects sought in the preparation phase (training and controlled diet) typically, designed to increase endurance level and increase strength [23]. In fact, in a previous study, Mascherini et al. [17] suggested that the shortening of the vector was associated with changes in hydration status and increases in body cell mass. In this study, the preparation phase could have increased the intracellular/extracellular water (ICW/ECW) ratio as can be seen from a higher PhA than that measured by Levi Micheli et al. [14] (8.0 ± 0.5° vs. 7.7 ± 0.6°). Indeed, PhA is positively associated with the ICW/ECW ratio in athletes [18,24]. Bioelectric data reflect the content of body fluids and the cellular health of the athlete and during the season, which change in response to training load and physical condition over the season [25]. In fact, the new tolerance ellipses proposed in this study differ significantly from those generated in the study by Levi Micheli et al. [14], in which bioimpedance measurements were collected in the first half of the competitive period. Furthermore, Micheli Levi et al. [14] reported that BIA data was collected over 5 months, from October to January 2009–2010, a period of time that may have generated vector changes in the athletes themselves. Our hypothesis is that the increase in workload (training) and official matches from August to October (about 6–8 matches played) or from August to January (16–17 matches played) could lead to fatigue and increased muscle turnover, as well as reduced muscle function which could result in a shift to the right of the biompedance vector. In fact, during the season, the reduction of the PhA could indicate a decreased muscle function as shown by Norman et al. [26] However, since we have not performed any muscle function tests, this hypothesis will have to be further investigated in future studies.

The reference ellipses proposed in the literature for athletes are population-specific. In addition to those for soccer players proposed by Levi Micheli [14], Campa and Toselli [15] measured male volleyball players in the second half of the in-season and showed specific BIA vector distribution in elite players in comparison to lower levels athletes. Subsequently, Giorgi et al. [16] provided bioelectrical impedance data of male road cyclists of varying performance levels, measured at the time of their optimal performance level and identified the 50, 75 and 95% tolerance ellipses for the road cyclists population, as well as for the high-performance road cyclists. In addition to these, there are also ellipses for healthy athletes built on more than 1000 male and 440 female athletes during the off-season period, therefore suitable for evaluating BIVA in the first phase of the competitive season [12].

The authors are also aware of the limitations of the study. Firstly, the subjects come from the same territory; therefore, the results obtained are not generalizable to all the soccer players around the world: a larger sample size is required even in different countries. The second is that no division by ethnicity of the players has been made in order to obtain a sample as large as possible: currently an international data collection is active that will allow us to investigate both these two limitations.

A strength of this study is in the specific time period in which the measurements were collected, not only in regard to the competitive level of the athletes, but above all for the time span in which BIA assessments were performed. In fact, BIA measurements were collected within a week, just before the start of the season, a period of time too short to generate vector changes between the players.

For the reasons mentioned above, future studies conducted with the aim of providing BIVA references for athletes should carry out the measurements according to the competitive phase for which they want to provide the new references. This is very significative given that vector changes occur during the different phases of the season in athletes, and bioelectrical values must be as informative and specific as possible, in order to obtain accurate monitoring of the body composition and physical condition of the athlete. This study demonstrates the importance of evaluating athletes on period-specific BIVA references, providing new tolerance ellipses for assessing body composition and cellular health before the start of the competitive season in elite soccer players.

## 5. Conclusions

Through BIVA, it is possible to evaluate body composition and the state of physical condition in the different phases of the competition in elite soccer players. This study provides specific BIVA references for the start of the season period, through which the physical condition achieved after the preparation micro cycle in soccer can be assessed.

## Figures and Tables

**Figure 1 jfmk-05-00073-f001:**
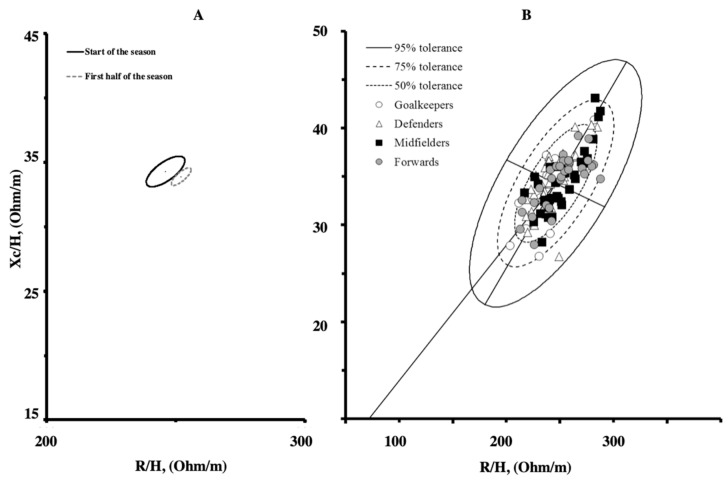
Mean impedance vectors with the 95% confidence ellipses for the soccer players measured at the start and at the first half of the competitive season [10] (Panel **A**). Scattergrams of the individual impedance vectors plotted on the new tolerance ellipses (Panel **B**).

**Table 1 jfmk-05-00073-t001:** Descriptive statistics for the soccer players according to playing position.

Variable	Goalkeepers *n* = (15)	Defenders *n* = (38)	Midfielders *n* = (38)	Forwards *n* = (40)	All *n* = (131)
Age (years)	24.2 ± 5.9	26.6 ± 4.8	25.0 ± 4.8	24.5 ± 3.9	25.1 ± 4.7
Weight (kg)	86.7 ± 5.4	80.6 ± 5.6	76.8 ± 5.6	77.6 ± 6.5	79.3 ± 6.6
Height (cm)	188.3 ± 3.5	185.1 ± 5.0	181.4 ± 5.2	181.8 ± 7.2	181.8 ± 7.2
BMI (kg/m^2^)	24.5 ± 1.0	23.5 ± 0.8	23.3 ± 1.0	23.5 ± 1.0	23.5 ± 1.0
R/H (ohm/m)	234.0 ± 18.1	242.9 ± 17.0	251.9 ± 18.9	254.6 ± 21.9	248.1 ± 20.3
Xc/H (ohm/m)	33.0 ± 3.9	34.4 ± 3.1	34.7 ± 3.2	35.4 ± 3.0	34.6 ± 3.3
PhA (degree)	8.0 ± 0.7	8.1 ± 0.5	7.8 ± 0.4	7.9 ± 0.4	8.0 ± 0.5

Abbreviations: BMI, body mass index; R/H, resistance standardized for height; Xc/H, reactance standardized for height; PhA, phase angle.

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
