# Peer review of "Bioimpedance Vector References Need to Be Period-Specific for Assessing Body Composition and Cellular Health in Elite Soccer Players: A Brief Report"

_jfmk, 2020, doi:10.3390/jfmk5040073_

Round 1

Reviewer 1 Report

review the attached document

Author Response

I have enjoyed reading your manuscript, which is interesting, rigorous, and adequate.

However, I am going to make some suggestions for improvement that I think can

contribute to increasing the quality of the manuscript.

The authors thank for the work done by the reviewer in order to improve their manuscript.

Replies to comments are highlighted in red below.

1) Abstract

 Line 22: The acronym BIVA should not appear directly in the text in the abstract.

BIVA has been previously define at line 16 and is an important abbreviation also as keyword

2) Introduction

 Generally very short, the background information found in the bibliography is

not sufficiently reported. It should be expanded considerably.

 Line 37-70, include any quotation that supports this idea.

 In the introduction a paragraph could be introduced that talks in greater depth

about::

o The importance of analyzing body composition in high-level sport.

o The connection between improvements in body composition and

athletic performance.

Thanks for your constructive comments. We implemented the introduction trying to respect the limits imposed by the short report format.

3) Materials and Methods (line 60 to 95). Some suggestions are proposed.

 Distribute in the following sections: Design and Participants; Instruments;

Process; Data analysis.

 Explain the process of selecting the sample of participants (in participants),

 Explain the inclusion and exclusion criteria to participate in this study (in

procedures).

 Specify if minor students are asked for permission from their parents or legal

guardians (in procedures).

Selection and inclusion / exclusion are reported at line 68. No minor soccer players were included in the study.

 Line 68. Detail that the study complies with the Ethical principles of the

Declaration of Helsinki in forcé (in participants, just after “by the Bioethics

Committee of the University of Milan (approval code: 1052019”).

Thank you for the suggestion. The correction has been made at line 74-75

4) Results

 They have been addressed correctly.

Thank you for the comment

5) Discussion

 I consider that the citations used are up-to-date, but very scarce. Only 14

studies have been referenced for this study.

 Another review is recommended, in order to compare these results with more

studies.

 You can also try to explain the findings by trying to establish causal

relationships between physical condition and sports performance.

6) Conclusions

 A paragraph on strengths and limitations of the study should be included.

 Describe future lines of research as possible improvements to this research.

We have implemented the discussion section by adding new and updated references

Strengths and limitation were already described at line150-159

Reviewer 2 Report

This reviewer congratulate with authors for the quality of this Brief Report.

Only minor formatting adjustments are required: 

Line 44: soccer players4, volleyballplayers5 and cyclist6. Add square brackets to the reference numbers

Line 63: was recruited, replace with were (131 players)

Line 135: matches from August to October (about 6-8 games played). Uniform the text using matches

Line 142: Campa and Toselli5. Add square brackets to the reference number

Lined 144: Giorgi et al6. Add square brackets to the reference number

Author Response

This reviewer congratulate with authors for the quality of this Brief Report.

The authors thank for the work done by the reviewer in order to improve their manuscript.

Replies to comments are highlighted in red below.

Only minor formatting adjustments are required: 

Line 44: soccer players4, volleyballplayers5 and cyclist6. Add square brackets to the reference numbers

Line 63: was recruited, replace with were (131 players)

Line 135: matches from August to October (about 6-8 games played). Uniform the text using matches

Line 142: Campa and Toselli5. Add square brackets to the reference number

Lined 144: Giorgi et al6. Add square brackets to the reference number

Thank you for the comment. All the correction requested has been done accordingly.

Round 2

Reviewer 1 Report

Although the introduction has been expanded slightly, the possibility of increasing this section should be considered, following the recommendations made in the first review.

Author Response

The authors thank the reviewer for the work done in order to improve our msnuscript.

The addition in the manuscript and the replies to comment are highlighted in blue.

Although the introduction has been expanded slightly, the possibility of increasing this section should be considered, following the recommendations made in the first review.

The introduction section has been improved with particular focus, as suggested, on:

- The importance of analyzing body composition in high-level sport.

- The connection between improvements in body composition and

athletic performance.

The sentence added is: 

"In sports, excess fat mass reduces endurance performance, while an increase in lean mass, especially muscle mass, is associated with an increase in power and strength [2]. Furthermore, the assessment of localized body composition allows the identification of differences in muscle mass and strength between areas of the body and may allow a reduction in the risk of injury (evaluation of contralateral limbs, agonist-antagonists) [3].

Body composition assessment should also be considered in sports involving weight categories, where athletes benefit from being placed in a lower weight category, in these cases any weight loss must therefore be monitored closely. Excessive training coupled with calorie restrictions can lead to excessive, unnecessary and dangerous weight loss. This weight loss in both women and men decreases performance, bone mineral density, muscle mass and is detrimental to health [4, 5]."